# A Multicriteria Model for the Assessment of Countries’ Environmental Performance

**DOI:** 10.3390/ijerph16162868

**Published:** 2019-08-10

**Authors:** Francisco Guijarro

**Affiliations:** Research Institut for Pure and Applied Mathematics, Universitat Politècnica de València, 46022 Valencia, Spain; fraguima@upvnet.upv.es; Tel.: +34-96-387-7000

**Keywords:** multicriteria environmental performance, goal programming, ranking, weighting

## Abstract

Countries are encouraged to integrate environmental performance metrics by covering the key value-drivers of sustainable development, such as environmental health and ecosystem vitality. The proper measurement of environmental trends provides a foundation for policymaking, which should be addressed by considering the multicriteria nature of the problem. This paper proposes a goal programming model for ranking countries according to the multidimensional nature of their environmental performance metrics by considering 10 issue categories and 24 performance indicators. The results will provide guidance to those countries that aspire to become leaders in environmental performance.

## 1. Introduction

The evidence for climate change and global warming is supported by extensive scientific research, while new reports and statistics are warning about its consequences all around the globe [1,2,3,4]. This situation will bring serious economic, political, and social problems in the immediate future and will significantly affect the survival of plants and animals on Earth [5,6,7]. This extreme challenge requires a global commitment involving all governments to limit global warming, stabilize CO_2_ emissions, preserve biodiversity, among others [8], to limit the effects of climate change. Reliable and objective indicators involving all the dimensions of environmental performance are needed to effectively combat its consequences.

The Environmental Performance Index (EPI) serves as an example of the measurement of high-priority environmental issues in different countries. The annual report ranks 180 countries on 24 performance indicators across ten issue categories covering environmental health and ecosystem vitality. These metrics provide national gauges of how close countries are to the established environmental policy goals [9].

A key element in the establishment of any composite index is the means by which appropriate weighting and aggregation procedures are selected. According to [9], the significance of weighting is twofold; it first refers to the explicit importance we give to every indicator in the composite index, and secondly it relates to the implicit importance of the indicators, considering the ‘trade-off’ between the pairs of indicators in the aggregation process.

Equal weighting is among the popular approaches [10,11]. Giving the same weight to all environmental indicators means that all the variables are worth the same in the composite index, but this could also disguise the absence of a statistical or an empirical basis [12]. Furthermore, correlation issues among indicators should be accounted for, otherwise some indicators can be positively aligned with others, so overestimating the relevance of the corresponding environmental dimension, while still others can be negatively correlated with others, thus undervaluing their weight in the composite index computation. Capturing synergies and trade-offs among the indicators can reveal crucial differences in scores and rankings [13].

Despite their vague definition, composite indicators have gained surprising popularity in all areas of research [11]. A Condorcet consistent nonlinear/noncompensatory mathematical aggregation rule for the construction of composite indicators is proposed in [14], under the assumption that the linear aggregation rule is not appropriate for building relevant composite indicators. In [15] it is argued that Condorcet rules present a lower probability of rank reversal than any other scoring method and are not compensatory. However, they also state that “a weak point is the high probability of the presence of cycles, whose solution normally implies ad hoc rules of thumb”. All possible preferences among the indicators are considered in [16]. Under each preference, authors develop a mathematical transformation to calculate the least and most favourable scores of each entity to formulate the lower and upper interval bounds. Composite indicators are constructed by means of an interval decision matrix.

A mathematical programming approach is proposed by [17] to construct a composite index, assuming that subjectivity in assigning weights to indicators is a critical issue in composite index computation. The proposed model is based on the Data Envelopment Analysis (DEA) from [18], and uses “two sets of weights that are most and least favourable for each entity to be evaluated and therefore could provide a more reasonable and encompassing composite index”. An analysis of DEA with different combinations of normalization, weighting, and aggregation methods is proposed by [19] for the assessment of an industrial case study on sustainability performance evaluation.

Following the mathematical programming proposals, a goal programming model is introduced by [20] for estimating the performance measure weights of firms by means of constrained regression. The model is underpinned by two extreme alternatives: the first is to calculate a consensus performance that reflects the majority trend of the single indicators, and the other is to calculate a performance that is biased towards the measures that show the most discrepancy with the rest. This approach was also applied to rank Spanish banks according to their financial statements [21], measure the social responsibility of European companies [22], and design a sustainable development goal index [23].

A further discussion of environmental index composition can be found in [11,24,25].

This paper aims to provide an objective and unifying method to weight different environmental dimensions by using a goal programming (GP) model, a well-established multicriteria technique. This methodology makes it possible to construct an environmental composite index by combining both positively and negatively-related indicators, which in the context of this study translates into models that allow simultaneous consideration of the different dimensions that make up countries’ environmental performance. Two extreme approaches are considered: The first prioritizes those environmental indicators that are aligned with the general trend of the dimensions that define the environmental behaviour of countries (the weighted goal programming model). The second favours those singular, conflicting environmental dimensions (the MINMAX goal programming model). A compromise alternative between these two extremes is chosen to elicit the countries’ weighting sensitivity and ranking variability: the extended goal programming model. The model allows countries to be ranked according to their environmental behaviour from a multicriteria perspective. The proposal gives a range of environmental performance values instead of a single crisp value and was applied to the data from the most recent Environmental Performance Index Report [9]. The results show that European countries occupy the highest positions, regardless of the goal programming approach used.

The rest of the paper is structured as follows. Section 2 introduces the database used in this research and the proposed goal programming models. Section 3 discuss the results obtained when applying the methodology to the data from Section 2 in which countries were ranked according to their environmental performance by objectively computed indicator weights. Our main conclusions are given in Section 4.

## 2. Materials and Methods

This section introduces the data set used in the research for the assessment of the environmental performance, and the proposed methodology to construct the multicriteria environmental performance index. Dataset is available at https://zenodo.org/record/3359779.

### 2.1. Data Source

The Environmental Performance Index (EPI) ranks 180 countries by different indicators, including environmental health and ecosystem vitality. The 24 performance indicators are grouped across ten issue categories, as shown in Table 1, which gives the relative weights of each policy objective, issue category and indicator in the “Weight” columns. According to [9], the indicators used in the report satisfy different criteria: relevance, performance orientation, established methodology, verification, completeness, and quality. The authors use logarithmic transformation to deal with skewness. The indicators are standardized on a 0–100 scale and then aggregated at each level of the proposed hierarchy. Indicator scores are aggregated into issue category scores, issue category scores into policy objective scores, and policy objective scores into final EPI scores. The authors state that “the weights used to calculate EPI scores represent just one possible structure, and we recognize that EPI users may favour different weights”. Our proposal is thus intended to create a methodology to objectively determine the weights of environmental indicators, considering their underlying interrelationships.

In the empirical Section 3 we give some descriptive statistics and the correlation matrix. The latter gives an idea of the relationships between certain indicators, which can distort the weight of environmental dimensions in EPI computation. As the proposed model needs a complete data set without missing values for all the indicators considered, the final sample includes 91 of the 180 countries initially included in the EPI report.

### 2.2. Basics on Goal Programming

Goal Programming (GP) was originally introduced by [26] to obtain constrained regressions estimates for an executive compensation formula. However, the term GP did not appear until the publication of *Management Models and Industrial Applications of Linear Programming* [27]. As pointed by [28], GP is based on the concept of satisfying objectives. In today’s complex organizations, decision makers (DMs) do not maximize a defined utility function. We must consider that “the conflicts of interest and the incompleteness of the available information make it almost impossible to build a reliable mathematical representation of DMs’ preferences. On the other hand, within this kind of decision environment DMs try to achieve a set of goals (or targets) as closely as possible” [28].

The objective of GP is the simultaneous optimization of different goals by minimizing deviations from predefined targets. In this regard, GP is aligned with some heuristic algorithms proposed in the Artificial Intelligence area: they cannot ensure the optimal solution but do provide one close to it. From a mathematical perspective, GP can be expressed as an optimization model that minimizes the deviation between the achievement of goals and their optimal aspiration levels. According to [22], its basic formulation can be expressed as Model (Equation 1).
(1)$$\begin{array}{ccc} \min & \sum_{j=1}^{m}|f_{j}\left(x\right)-g_{j}| & \\ & s.t. & \\ & x\in F & F\;\mathrm{is}\;a\;\mathrm{feasible}\;\mathrm{set} \end{array}$$
where x is a vector of decision variables, $f_{j}\left(x\right)$ is a linear function of the *j*-th goal, and $g_{j}$ is its aspiration level.

Model (Equation 1) can be easily transformed into a linear model by introducing positive and negative deviations (Model (Equation 2)):(2)$$\begin{array}{ccc} \min & \sum_{j=1}^{m}\left(d_{j}^{-}+d_{j}^{+}\right) & \\ & s.t. & \\ & f_{j}\left(x\right)+d_{j}^{-}-d_{j}^{+}=g_{j} & j\in \{1,\ldots ,m\}\\ & x\in F,d^{-}\geq 0,d^{+}\geq 0 & \end{array}$$
where $d_{j}^{-}$, $d_{j}^{+}$ are negative and positive deviations from target value of *j*-th goal.

### 2.3. Measuring Environmental Performance through a Goal Programming Model

We propose to measure countries’ overall environmental performance by means of a GP model in order to obtain a single measure of Multicriteria Environmental Performance (MEP) as an aggregation of the different indicators that measure their environmental behaviour.

The present study is based on the aggregating procedure proposed by [29], in which different GP models are used to aggregate the preferences of social groups. This approach has been successfully applied to ranking banks [21], firms [20], and measuring firms’ social responsibility [22]. Depending on the norm used, the solution obtained can be interpreted either as one in which the consensus between all the indicators is maximized (penalizing the more conflicting indicators in favour of those that are more representative of the majority trend) or as one in which preference is given to the most conflicting indicators (thereby penalizing the measures that share the most information with the rest) [20]. In the first case, the absolute difference between the multicriteria environmental performance and the single indicators is minimized (norm $L_{1}$); in the second case, the greatest difference between the multicriteria performance and the single indicators (norm $L_{\infty }$) is minimized. In the following, we introduce the models used throughout this subsection.

Let *n* represent the number of countries and *m* the number of environmental indicators. The composite weights, w, are computed by multiplying the corresponding relative weights. Then, the MEP of any country *i* ($MEP_{i}$) can be computed by composing the abovementioned *m* weights with the environmental indicators measured for each country i∈{1,…,n}:(3)$$\begin{array}{cc} MEP_{i}=w^{t}\times {\mathrm{epi}}_{i}=\sum_{j=1}^{m}w_{j}epi_{ij} & i\in \{1,\ldots ,n\} \end{array}$$

We propose to compute the Multicriteria Environmental Performance by objectively determining the weights of each indicator through a GP model.

The first GP Model (4) calculates the MEP by maximizing the similarity between this measure and the other environmental indicators considered as inputs to the model. This model is known as the weighted goal programming (WGP) model and uses the $L_{1}$ norm.

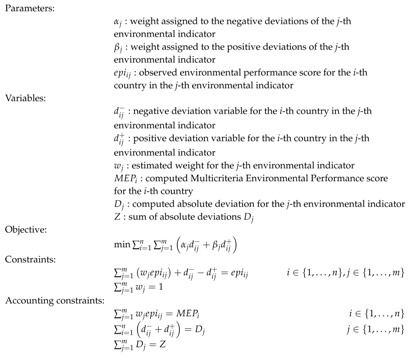
(4)
where all variables are supposed to be positive. We also assume $\alpha_{j}=1$ if $d_{ij}^{-}$ is unwanted, otherwise $\alpha_{j}=0$; $\beta_{j}=1$ if $d_{ij}^{+}$ is unwanted, otherwise $\beta_{j}=0$. There are situations where we want to penalize the overachievement of the goals, but not its underachievement; and vice versa. For example, if we are dealing with a cost related variable, we may be interested in controlling the expenses by penalizing the overachievement of the cost goal. However, we may allow for the underachievement of its value. An example can be found in [22]. The variable $w_{j}$ is the computed weight for the *j*-th environmental indicator; $d_{ij}^{-}$ and $d_{ij}^{+}$ are negative and positive deviations, respectively. The latter variables quantify the difference by excess (deficiency) between the environmental indicator of the *i*-th country in the *j*-th indicator and the estimated MEP. $D_{j}$ accounts for the disagreement between the *j*-th indicator and the computed MEP. Hence, $D_{j}$ quantifies the difference between countries in the *j*-th indicator with respect to the estimated multicriteria environmental performance. *Z* is the sum of the overall disagreement. We must point out that the sum of $D_{j}$ may be different from the value obtained in the objective function unless we assume $\alpha_{j}=\beta_{j}=1$. Different combinations of values for $\alpha_{j}$ and $\beta_{j}$ serve to perform a trade-off between negative and positive deviations. This would translate into changes regarding the computed weights w.

Constraint $\sum_{j=1}^{m}\left(w_{j}epi_{ij}\right)+d_{ij}^{-}-d_{ij}^{+}=epi_{ij}$ is split into a total of n×m equations. For each country *i*, as many equations are created as the indicators considered to measure environmental performance, i.e., *m* equations. In each of these equations the country’s estimated multicriteria environmental performance is compared to its performance in the *j*-th indicator. The estimated multicriteria environmental performance is computed as the composition of environmental indicators $\sum_{j=1}^{m}\left(w_{j}epi_{ij}\right)$, and is summarized as $MEP_{i}$ in constraint $\sum_{j=1}^{m}w_{j}epi_{ij}=MEP_{i}$. This value is unique for each country, obtained from the estimated weights $w_{j}$. The difference between this value and each of the different *m* values of the single indicators, $epi_{ij}$, is computed by the deviation variables: $d_{ij}^{-}$ and $d_{ij}^{+}$. That is, $d_{ij}^{-}-d_{ij}^{+}=epi_{ij}-\sum_{j=1}^{m}w_{j}epi_{ij}=epi_{ij}-MEP_{i}$.

Constraint $\sum_{j=1}^{m}w_{j}=1$ determines that the sum of the weights must be one. The last 3 constraints in Model (4) are accounting constraints. Constraint $\sum_{i=1}^{n}\left(d_{ij}^{-}+d_{ij}^{+}\right)=D_{j}$ computes the value of $D_{j}$ for each indicator measure, as the sum of the absolute differences between the estimated multicriteria environmental performance and the original indicators. It should be noted that a high $D_{j}$ value indicates a high degree of disagreement between the *j*-th environmental indicator and the estimated multicriteria environmental performance. On the other hand, low values indicate that the countries’ behaviour in that indicator is very close to the global multicriteria environmental performance. The sum of all disagreements is computed in constraint $\sum_{j=1}^{m}D_{j}=Z$ and coincides with the value of the objective function. In this way, a model with a low *Z* value indicates that the computed multicriteria environmental performance is in line with all the single indicators, and a high value means that there are large differences between the two values. This situation will occur when single indicators are very dissimilar to each other.

It should be noted that the primary objective is to achieve a unique measure of the environmental performance that is in line with the different single indicators used in the analysis. However, this may be more complicated when these indicators are in conflict with each other, so that a high value in one can imply a low value in another. If countries can improve one environmental dimension without worsening the rest, it can be assumed that the multicriteria estimate is aligned with the set of environmental dimensions.

As weights $w_{j}$ are calculated objectively, no subjective opinion is considered regarding the importance of each environmental indicator. Note that all deviation variables have been equally weighted in the objective function, which should not be understood as meaning that all the variables are equally important.

The norm $L_{\infty }$ is obtained by the MINMAX GP Model (5), in which *D* represents the maximum deviation between the computed MEP and the single environmental indicators.

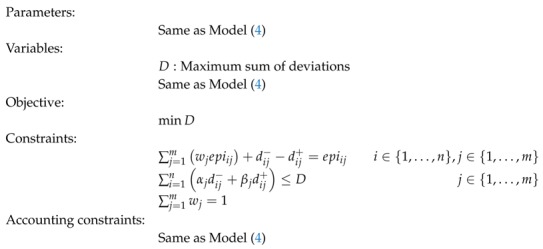
(5)

Two differences must be emphasized regarding the previous model. The first is the objective function, where the maximum deviation *D* between the MEP and the individual indicators is minimized. The second is the new constraint $\sum_{i=1}^{n}\left(\alpha_{j}d_{ij}^{-}+\beta_{j}d_{ij}^{+}\right)\leq D$, which calculates the value of *D* as the supremum of the sum of deviations for each indicator *j*. The rest of the constraints remain the same as in the WGP Model (4). Both models represent extreme cases. The WGP model gives more importance to the general consensus between environmental indicators, while the MINMAX model prioritizes the conflicting indicators.

The parametric extended GP model seeks a compromise between these two extreme approaches [30]. With the extended GP Model (6), decision makers obtain alternative compromise solutions according to the value they assign to the parameter λ [20]. This broadens the range of possibilities when they have to decide what solution is the best suited to the environmental indicators and most representative of them. It can be observed in (6) how if λ=1, the same solution is obtained as in Model (4), whereas in the case of λ=0, the solution coincides with that of Model (5):

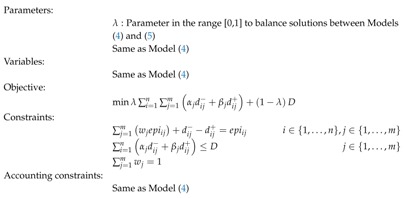
(6)

For a further discussion on goal programming and its variants we refer to [31,32,33].

## 3. Empirical Results

### 3.1. Descriptive Statistics

Table 2 shows some descriptive statistics for the environmental indicators included in the dataset. As we have previously mentioned, all variables are standardized on a scale of 0–100. It can be seen that both the mean and median of most variables are over 50. The reason is that a high number of countries concentrate on the above-mean area for most variables. The distribution is skewed to the left on the basis of the reported skewness coefficients.

The reported correlation coefficients are low (Figure 1). However, the correlation coefficients reveal synergies and trade-offs between some environmental indicators. The most significant correlation is between drinking water (UWD) and wastewater treatment (WWT). Drinking water is measured as the proportion of a country’s population exposed to health risks from their access to drinking water, and wastewater treatment is defined as the percentage of wastewater that undergoes at least primary treatment in each country, normalized by the proportion of the population connected to a municipal wastewater collection system. Hence, the negative correlation coefficient (-0.49) indicates a trade-off between these two variables. Despite the reported relation, these indicators are classified in different issue categories and policy objectives. An example of synergies between indicators is the one given by the national biome protection (TBN) indicators and PM_2.5_ Exceedance (PMW). The reported correlation coefficient is +0.48, thus confirming a positive relationship between the percentage of biomes in protected areas and the proportion of the population exposed to ambient PM_2.5_ concentrations in excess of World Health Organization (WHO) thresholds. As in the previous example, these environmental indicators are grouped into different issue categories and policy objectives.

### 3.2. Assessing Multicriteria Environmental Performance

This Section analyzes the results obtained by applying Model (6) to the database described in the previous section. The extended version of the GP model was used because it embeds the WGP and MINMAX models, so that different MEPs were obtained according to whether some indicators are more strongly promoted than others, or whether a greater weight is given to the indicators that converge with the mean behaviour. Model (6) was solved for λ values between 0 and 1, increasing in steps of 0.01, so that the model was run 101 times.

For the sake of simplicity, Table 3 only reports the solution of Model (6) for some representative λ values. The first column, λ=0, is from Model (5), which prioritizes the most conflicting environmental indicators, or those least correlated with the majority trend. As several indicators have zero weight they do not contribute to the computation of the multicriteria environmental performance index. Despite zero weights are usual in GP models [20,21,22], doing away with criteria is not common in multicriteria decision making. The reason why several criteria are excluded by the GP model is the correlation structure between some criteria. For example, the variable SHI is excluded in all models regardless of the λ value. A stepwise regression explaining the SHI variable gets an adjusted R-Square of 31.5%, which corresponds to a multiple correlation coefficient of 0.613. Hence, other variables in the sample provide most of the information included in the SHI variable and the GP model discards this variable once those criteria are considered. The most significant indicators are PM_2.5_ Exposure ($w_{PME}=0.291$), CO_2_ Emissions – Power ($w_{DPT}=0.144$) and N_2_O Emissions ($w_{DNT}=0.137$). The distance between the computed multicriteria environmental performance and the original indicators can be seen to be constant: 1980. This translates into a multicriteria indicator which is equidistant from all the single indicators. Obviously, the maximum distance *D* equals the same value. On the opposite side, the last column, with λ=1, is from Model (4), which computes a multicriteria environmental performance index which minimizes the sum of absolute distances to each indicator. The *Z* value is 45,010, which is significantly lower than the value obtained for λ=0 (47,519). Of the weights associated with each single indicator, the most influential indicator is household solid fuels ($w_{HAD}=0.128$), followed by wastewater treatment ($w_{WWT}=0.101$). The weights are more balanced under this approach, and only 4 indicators get a zero value. There are significant differences in the indicators’ distance from the computed multicriteria performance, with the most similar being wastewater treatment, with a distance$D_{WWT}=1409.2$, whilst the least similar is PM_2.5_ exposure ($D_{PME}=2358.3$). This result is not surprising considering that WWT has the second largest weight in computing multicriteria environmental performance, and PME is the most important indicator in the λ=0 Model. There is also a trade-off between *D* and *Z* in the GP models, as the GP model prioritizes one variable over another, according to the λ value.

It may be easier to draw conclusions on the environmental dimensions and their weights in the GP models by analyzing the issue categories instead of working on the original environmental indicators. Figure 2 shows the weights obtained by the reported 10 issue categories in the GP model. The relative weight per category was computed by adding the weights of the corresponding indicators. For example, air quality (AIR) weight is the sum of the weights for household solid fuel (HAD), PM_2.5_ exposure (PME), and PM_2.5_ exceedance (PMW). It can be seen that some categories have a relatively low value, whatever their λ value.

Although the Forests (FOR) issue category has a weight of 6% in the EPI report [9], its maximum value throughout the entire λ range is 1.9%. Other categories with a low rate in the GP model include: Fisheries (FSH), with a weight of 6% in the EPI report and a maximum weight of 2.4% in the GP model; Agriculture (AGR) (3% and 3.8%), Water & Sanitation (H2O) (12% and 4.9%), and Heavy Metals (HMT) (2% and 5.3%). On the opposite side, the most heavily weighted category is Climate & Energy (CCE), with a maximum weight of 45.6% in the GP model and 18% in the EPI report, and Air Quality (AIR) with a maximum value of 33.1% and an EPI value of 26%. It can therefore be concluded that both the EPI report and the GP methodology identify the most and least important issue categories in a similar way, but the interesting point about the GP model is that its weights depend on the lambda value, so that the relative importance of each category can be computed as a range of values instead of a single crisp value. For example, the AIR category weight varies between 21.9% and 33.1%, in such a way that the relative importance of the environmental dimensions varies with the approach adopted by the decision maker (i.e., prioritize the consensus between all the indicators or give preference to the most conflicting indicators).

Because of the different weights obtained by the environmental indicators for the λ value, the environmental performance ranking of the countries can also differ. Figure 3 contains a boxplot with the multicriteria environmental performance index obtained by each country. The model generated 101 different MEP values per country for λ values ranging from 0 to 1 increased by 0.01. The countries are shown in descending order according to their median MEP.

It can be seen that certain countries occupy the top positions regardless of the λ approach used. The top 10 in the ranking are mostly in Europe: France, Sweden, Italy, Germany, Belgium, United Kingdom, Lithuania, Denmark, and Spain, with Israel as the only non-European country, which clearly shows the importance given to environmental issues in Europe. There is a similar geographical pattern at the bottom of the distribution. The least favourable positions are occupied by Asian and African countries: Pakistan, India, Kenya, Brunei Darussalam, Republic of Congo, Sudan, Bangladesh, and Senegal. We can thus conclude that environmental performance strongly depends on geographical position, obviously without ruling out the significant relationship between this factor and economic development, as pointed out in [34].

Another interesting point outlined in Figure 3 is the MEP dispersion. Some countries present a narrow range of MEP values, thus indicating that their environmental performance is closely independent on the approach considered in the goal programming model. For example, Lithuania is in a very good and stable position, with a maximum MEP of 83.36 and minimum of 75.84. On the other hand, Cameroon is one of the lowest environmentally ranked, regardless of the λ value, with a maximum MEP of 25.83 and a minimum of 23.08. Some countries have a wider range of MEP values; Denmark has an interquantile range between 81.08 and 68.42. Finally, the dispersion, measured as the difference between maximum and minimum values, is even greater: 84.15−43.82=40.33. To sum up, the proposed model provides a flexible assessment of multicriteria environmental performance instead of working with a single crisp value.

Figure 3 also includes the country ranking obtained by applying the weights from Table 1 [9], where scores have been normalized in the range [0,1]. Each country is represented by a single point, in contrast to the GP ranking where λ parameter enables to obtain a non-crisp value of the environmental performance. The best and worst performer countries in the GP methodology are also consistently ranked in terms of the EPI report. This translates in European countries being in the top of the EPI ranking. However, we can observe a higher variability in those countries located in the middle of the ranking. Finland or Australia were middle performers under the GP methodology, but both of them are highly ranked in the EPI report.

An important conclusion is that most points are on the right side of the GP scores (i.e., on the right side of the boxplot). This means that most countries obtain a high score in the EPI report (58 out of 91 countries are above 50), and it is difficult to draw significant differences between some countries regarding its environmental performance. We have reported that 50% of countries get an EPI score between 39.7 and 71.8, and this makes difficult to highlight significant differences in middle performer countries. In contrast, the ranking obtained through the GP methodology is more balanced, with approximately the same number of countries above and below the level of 50.

## 4. Conclusions

A recent survey reported that UK citizens were more worried about climate change than the economy, crime, and immigration. This trend is becoming increasingly widespread, especially in the wealthier nations. As a result of the multiple dimensions involved in countries’ environmental performance, many can be outstanding in a certain environment indicator but deficient in another. Another related concern is how policy objectives, issue categories, and indicators are weighted in the EPI performance index, which makes it difficult to objectively rank countries according to their environmental performance.

This paper proposes a multicriteria approach to objectively estimate an environmental performance index through goal programming, considering synergies and trade-offs between the different environmental dimensions. Goal programming permits two extreme approaches to be used: (a) either to prioritize the environmental indicators that are aligned with the general trend of other dimensions, defining the country’s environmental behaviour (the WGP model), or (b) to favour the singular, conflicting environmental dimensions (MINMAX GP model). The extended GP model gives a compromise between these two extremes. This model identifies the most important environmental dimensions, regardless of the approach considered. In a similar way, the results show that most countries’ ranking remains steady, regardless of the goal programming perspective. The high degree of dispersion in multicriteria environmental performance shows that while some countries behave well in some indicators they perform poorly in others, showing that their environmental ranking fluctuates according to the weights given to the influential indicators.

We can conclude that it seems to be a universal desire to aggregate in order to rank countries by their environmental performance. As a practical matter, doing this has fairly little value other than to make the citizens of wealthy, developed countries feel good. In reality, those citizens are largely responsible for the poor environmental performance of many developing countries through their consumption of imports produced in those countries.

## Figures and Tables

**Figure 1 ijerph-16-02868-f001:**
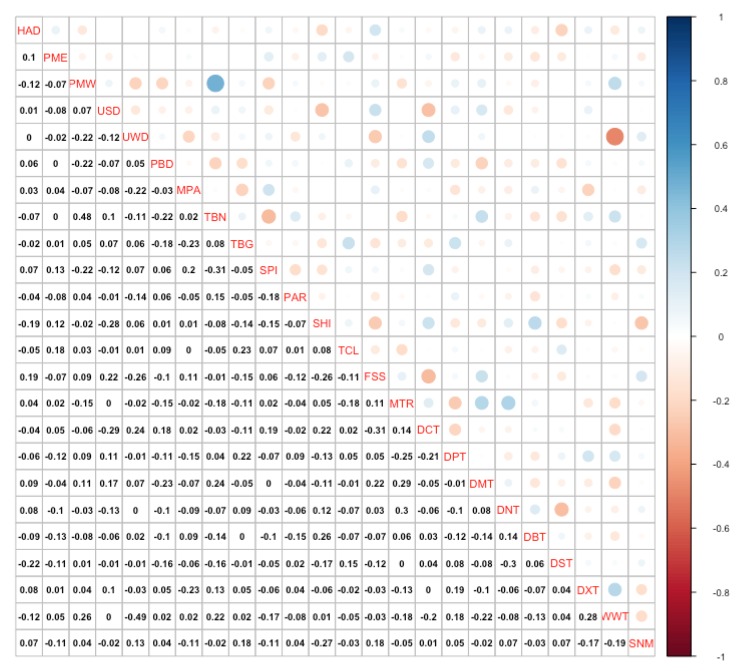
Correlation coefficients for the environmental indicators.

**Figure 2 ijerph-16-02868-f002:**
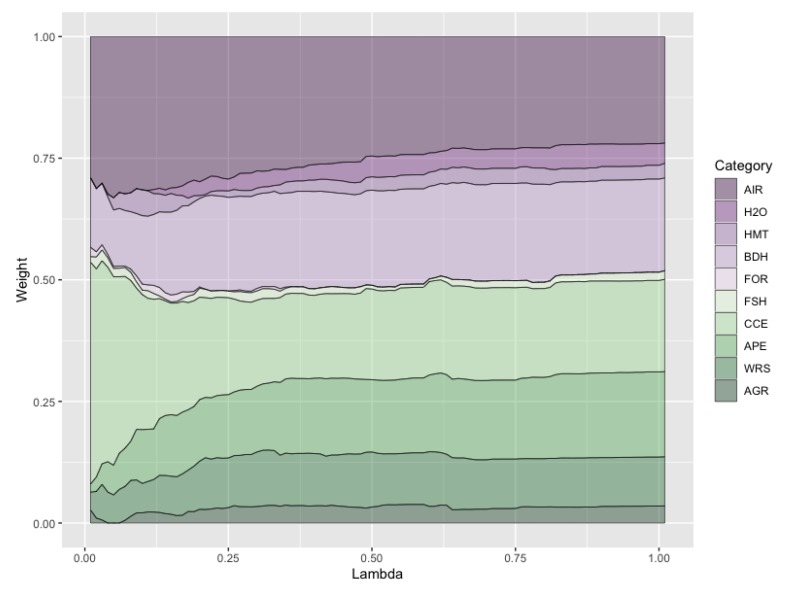
Aggregated weights for the issue categories of the environmental performance.

**Figure 3 ijerph-16-02868-f003:**
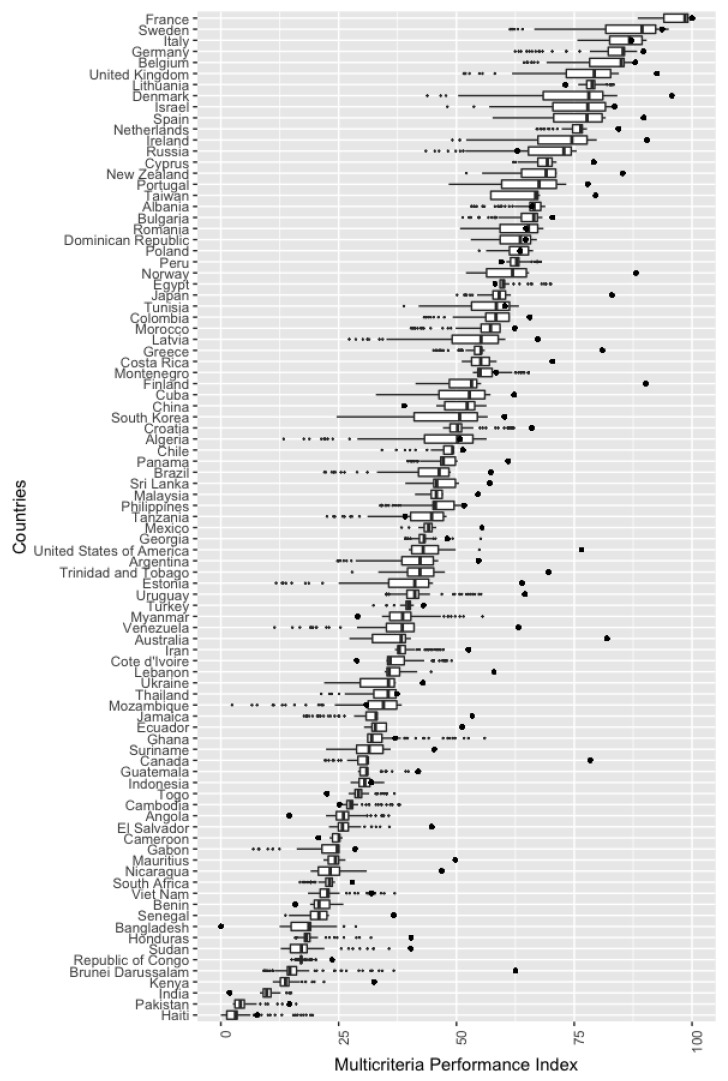
Multicriteria Environmental Performance (EPI) values according to the λ values used in the goal programming model. The EPI score from [9] is represented by points.

**Table 1 ijerph-16-02868-t001:** Policy objectives, issue categories, indicators, and weights from the Environmental Performance Index.

Policy Objective	TLA	Weight	Issue Category	TLA	Weight	Indicator	TLA	Weight	*w*
Environmental Health	HLT	40%	Air Quality	AIR	65%	Household Solid Fuels	HAD	40%	10.4%
						PM_2.5_ Exposure	PME	30%	7.8%
						PM_2.5_ Exceedance	PMW	30%	7.8%
			Water & Sanitation	H2O	30%	Drinking Water	UWD	50%	6.0%
						Sanitation	USD	50%	6.0%
			Heavy Metals	HMT	5%	Lead Exposure	PBD	100%	2.0%
Ecosystem Vitality	ECO	60%	Biodiversity & Habitat	BDH	25%	Marine Protected Areas	MPA	20%	3.0%
						Biome Protection (National)	TBN	20%	3.0%
						Biome Protection (Global)	TBG	20%	3.0%
						Species Protection Index	SPI	20%	3.0%
						Representativeness Index	PAR	10%	1.5%
						Species Habitat Index	SHI	10%	1.5%
			Forests	FOR	10%	Tree Cover Loss	TCL	100%	6.0%
			Fisheries	FSH	10%	Fish Stock Status	FSS	50%	3.0%
						Regional Marine Trophic Index	MTR	50%	3.0%
			Climate & Energy	CCE	30%	CO_2_ Emissions – Total	DCT	50%	9.0%
						CO_2_ Emissions – Power	DPT	20%	3.6%
						Methane Emissions	DMT	20%	3.6%
						N_2_O Emissions	DNT	5%	0.9%
						Black Carbon Emissions	DBT	5%	0.9%
			Air Pollution	APE	10%	SO_2_ Emissions	DST	50%	3.0%
						NO_X_ Emissions	DXT	50%	3.0%
			Water Resources	WRS	10%	Wastewater Treatment	WWT	100%	6.0%
			Agriculture	AGR	5%	Sustainable Nitrogen Management	SNM	100%	3.0%

**Table 2 ijerph-16-02868-t002:** Descriptive statistics of the environmental performance indicators.

TLA	Mean	Sd	Median	Skewness	Kurtosis
HAD	53.95	27.16	56.17	−0.09	−1.08
PME	59.04	33.55	63.43	−0.30	−1.26
PMW	58.85	30.47	58.83	−0.12	−1.21
USD	51.67	29.88	49.66	0.01	−1.07
UWD	61.46	26.74	63.79	−0.49	−0.51
PBD	68.25	26.44	67.57	−0.46	−0.81
MPA	60.58	26.34	60.13	−0.17	−0.82
TBN	67.74	29.60	67.77	−0.47	−0.94
TBG	61.72	27.78	63.72	−0.41	−0.82
SPI	48.98	32.27	44.22	0.21	−1.19
PAR	52.43	28.78	51.34	−0.01	−1.21
SHI	70.36	27.16	76.51	−0.59	−0.90
TCL	59.19	29.15	59.69	−0.14	−1.21
FSS	57.92	25.68	59.14	−0.14	−0.68
MTR	55.88	28.47	60.36	−0.13	−1.14
DCT	69.43	29.33	77.72	−0.69	−0.66
DPT	58.24	27.00	54.63	−0.06	−1.25
DMT	60.09	27.75	62.54	−0.18	−1.00
DNT	57.32	26.81	53.22	0.09	−1.06
DBT	62.89	29.98	64.31	−0.31	−1.22
DST	66.77	29.17	71.26	−0.59	−0.80
DXT	54.90	28.39	53.08	0.03	−0.98
WWT	60.22	29.92	61.03	−0.33	−1.05
SNM	57.44	28.65	57.15	−0.18	−0.93

**Table 3 ijerph-16-02868-t003:** Results obtained of the goal programming model for different λ values.

λ	0.0	0.1	0.2	0.3	0.4	0.5	0.6	0.7	0.8	0.9	1.0
$w_{HAD}$	0.000	0.080	0.102	0.099	0.106	0.122	0.124	0.126	0.129	0.129	0.128
$w_{PME}$	0.291	0.219	0.146	0.128	0.114	0.084	0.073	0.055	0.047	0.041	0.040
$w_{PMW}$	0.000	0.017	0.045	0.050	0.042	0.041	0.041	0.050	0.052	0.050	0.051
$w_{UWD}$	0.000	0.001	0.032	0.032	0.027	0.016	0.007	0.007	0.004	0.005	0.000
$w_{USD}$	0.000	0.000	0.000	0.000	0.004	0.027	0.031	0.033	0.040	0.041	0.042
$w_{PBD}$	0.000	0.052	0.003	0.013	0.024	0.028	0.029	0.031	0.031	0.028	0.030
$w_{MPA}$	0.030	0.028	0.024	0.023	0.036	0.044	0.045	0.045	0.048	0.047	0.047
$w_{TBN}$	0.000	0.014	0.047	0.062	0.049	0.026	0.015	0.027	0.037	0.037	0.039
$w_{TBG}$	0.000	0.015	0.015	0.003	0.022	0.040	0.053	0.043	0.033	0.036	0.033
$w_{SPI}$	0.034	0.024	0.045	0.040	0.032	0.031	0.028	0.032	0.029	0.023	0.025
$w_{PAR}$	0.079	0.060	0.061	0.065	0.061	0.056	0.048	0.053	0.051	0.048	0.046
$w_{SHI}$	0.000	0.000	0.000	0.000	0.000	0.000	0.000	0.000	0.000	0.000	0.000
$w_{TCL}$	0.019	0.012	0.000	0.004	0.000	0.000	0.000	0.000	0.000	0.000	0.000
$w_{FSS}$	0.011	0.008	0.000	0.000	0.000	0.000	0.000	0.000	0.000	0.000	0.000
$w_{MTR}$	0.000	0.007	0.017	0.020	0.013	0.007	0.006	0.015	0.014	0.016	0.018
$w_{DCT}$	0.071	0.043	0.040	0.039	0.039	0.041	0.044	0.041	0.043	0.044	0.045
$w_{DPT}$	0.144	0.079	0.031	0.014	0.019	0.024	0.027	0.024	0.021	0.024	0.022
$w_{DMT}$	0.042	0.015	0.045	0.042	0.070	0.057	0.058	0.057	0.054	0.061	0.057
$w_{DNT}$	0.137	0.117	0.057	0.042	0.015	0.024	0.029	0.027	0.027	0.022	0.025
$w_{DBT}$	0.062	0.015	0.033	0.039	0.030	0.038	0.034	0.041	0.038	0.038	0.040
$w_{DST}$	0.000	0.068	0.056	0.059	0.066	0.072	0.084	0.081	0.084	0.090	0.093
$w_{DXT}$	0.017	0.039	0.068	0.077	0.090	0.078	0.076	0.081	0.084	0.084	0.082
$w_{WWT}$	0.037	0.063	0.106	0.114	0.106	0.109	0.112	0.102	0.100	0.100	0.101
$w_{SNM}$	0.027	0.023	0.028	0.035	0.036	0.034	0.035	0.030	0.033	0.034	0.035
$D_{HAD}$	1980.0	1780.4	1694.4	1687.3	1671.4	1639.6	1638.4	1635.8	1630.7	1626.8	1631.2
$D_{PME}$	1980.0	2030.8	2099.5	2130.4	2172.6	2246.4	2278.1	2317.6	2331.7	2351.7	2358.3
$D_{PMW}$	1980.0	1903.1	1846.9	1839.6	1861.4	1877.0	1886.4	1883.2	1879.5	1887.7	1890.6
$D_{UWD}$	1980.0	1796.1	1773.0	1760.8	1775.1	1757.9	1762.8	1768.0	1764.8	1766.0	1772.3
$D_{USD}$	1980.0	1737.6	1706.6	1694.2	1690.2	1655.7	1654.8	1656.7	1648.3	1647.0	1651.1
$D_{PBD}$	1980.0	1830.8	1868.0	1837.3	1820.9	1802.1	1797.4	1791.0	1788.5	1789.5	1785.6
$D_{MPA}$	1980.0	1974.8	1935.5	1934.5	1915.1	1921.9	1926.1	1933.8	1933.5	1940.6	1941.3
$D_{TBN}$	1980.0	1972.8	1976.9	2000.4	2000.0	2035.5	2046.9	2048.7	2056.4	2063.3	2064.1
$D_{TBG}$	1980.0	1949.9	1970.9	1995.8	1989.7	2016.3	2020.6	2026.3	2034.3	2040.9	2041.2
$D_{SPI}$	1980.0	1992.2	1957.6	1978.1	1984.4	1981.1	1985.0	1972.3	1978.2	1988.0	1984.2
$D_{PAR}$	1980.0	1915.0	1890.6	1879.7	1864.9	1861.5	1865.8	1853.2	1855.0	1860.3	1863.2
$D_{SHI}$	1980.0	1801.0	1827.3	1829.7	1835.2	1848.7	1856.0	1865.5	1869.3	1878.1	1878.0
$D_{TCL}$	1980.0	1974.8	1950.5	1944.5	1962.8	1968.9	1973.2	1970.5	1973.8	1978.9	1978.5
$D_{FSS}$	1980.0	2030.8	2099.5	2114.5	2120.1	2117.6	2114.9	2107.3	2109.5	2111.8	2109.9
$D_{MTR}$	1980.0	1925.8	1912.4	1910.9	1917.1	1919.0	1917.6	1900.5	1903.1	1897.1	1892.7
$D_{DCT}$	1980.0	2030.8	2024.0	2034.0	2020.4	2004.8	1991.5	1991.0	1990.1	1985.4	1981.1
$D_{DPT}$	1980.0	2030.8	2099.5	2115.9	2107.2	2073.9	2064.3	2061.3	2062.6	2057.3	2057.2
$D_{DMT}$	1980.0	1985.9	1911.8	1906.3	1874.5	1874.0	1864.5	1854.2	1853.5	1840.7	1841.9
$D_{DNT}$	1980.0	2030.8	2099.5	2130.4	2164.7	2151.8	2145.6	2135.0	2135.8	2140.8	2138.7
$D_{DBT}$	1980.0	1962.3	1894.8	1863.4	1846.7	1814.5	1801.6	1782.4	1778.7	1761.6	1759.0
$D_{DST}$	1980.0	1736.2	1672.5	1644.5	1614.9	1591.5	1568.1	1568.2	1560.9	1546.6	1543.8
$D_{DXT}$	1980.0	1857.8	1765.0	1719.0	1693.5	1684.0	1671.2	1659.1	1650.4	1638.0	1635.7
$D_{WWT}$	1980.0	1579.3	1472.0	1431.8	1412.6	1402.7	1398.1	1416.2	1414.2	1408.6	1409.2
$D_{SNM}$	1980.0	1919.5	1880.9	1853.2	1840.7	1822.0	1813.4	1820.8	1811.7	1803.6	1801.2
*Z*	47,519	45,749	45,330	45,236	45,156	45,068	45,042	45,019	45,014	45,010	45,010
*D*	1980.0	2030.8	2099.5	2130.4	2172.6	2246.4	2278.1	2317.6	2331.7	2351.7	2358.3

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
