# Peer review of "A Multicriteria Model for the Assessment of Countries’ Environmental Performance"

_ijerph, 2019, doi:10.3390/ijerph16162868_

Round 1
Reviewer 1 Report
p.p1 {margin: 0.0px 0.0px 0.0px 0.0px; line-height: 15.0px; font: 13.0px Arial; color: #0000ff} p.p2 {margin: 0.0px 0.0px 0.0px 0.0px; line-height: 15.0px; font: 13.0px Arial; color: #0000ff; min-height: 15.0px}This article applies a known technique, goal programming, to a dataset specific to the journals aims and scope. The methods are sound, but their presentation could use a little improvement, as the topic is an important one and thus might attract readers outside the field of multicriteria analysis. The results also require some further work, as I now demonstrate.
As usual with articles whose main novelty lies in exploring a particular dataset, the aspect of discussing results gains major relevance, since otherwise the article’s contribution to the field is only a minor one. Comparison with other environmental indicators in the field is thus required. The discussion need not be lengthy, but some relevant comments are definitely required, so that the plus-values of the GP methodology can be highlighted. A suggestion is to compare, in some meaningful manner, the country rankings of the present article (fig.3) to e.g. those obtained by applying table 1 weights to the same dataset (also, are there any more articles that use table 1 indicators and data? Maybe recheck literature?). A critical analysis of the pros and cons of the proposed GP indicator would also be appreciated. Some compairson GP/EPI is done in L.259-272, but this is insufficient as it only deals with weights, not rankings. Note: this issue is the reason why the review form “Are the conclusions supported by the results?” was answered with “must be improved”.
Other major comments
In models (4-6) all variables and parameters should be explicitly indicated and their type mentioned (integer, binary, real, range, etc.). This is partly is done after the models are presented, but it would make the formulation more readable if preliminary information concerning the variables and parameters is discriminated beforehand. A suggestion is to follow a structure such as e.g. parameters/variables/objective/constraints, which is common when laying out linear programming models.
The role of alpha_j and beta_j in (4-6) should be better explained. “Unwanted” is not enough to understand why they feature in the model. Also, were any alpha/beta = 0 used in the calculations? If so, which one and why?
Again in models (4-6), some expressions are redundant. Indeed, the last three lines of these models are just bookkeeping constraints and do not participate in the calculations. They should be set aside from the model definitions or shown in a way similar to what was done in reference [20].
L.172 Values of the expression on line 6 of (4) does NOT coincide with the value of the objective function unless alpha_j and beta_j are 1 for all j. This should be clarified.
L. 234 The GP sets some weights to (exactly) zero. I have generated and ran some random GP models and verified that zero weights is actually a relatively common occurrence. While this is perfectly fine mathematically, doing away with criteria is not common in multicriteria decision making. Some comment on this particular point is needed. Perhaps correlations play a role here?
Minor comments
L.40 and below. Sentences should not start with “[15]” or with a number. Instead of “[15] argues that...” The following form is suggested “In [15] it is argued that...” The same applies to other iterations of the problem during the introduction.
L.54 “introduce” -> “introduces”
L.86 A download link to the dataset would be nice.
L.225 Figure 1. Some of the numbers are unperceptible due to light color. Please improve the figure for readability.
L.274-277 The paragraph is too long and repetitive. It should be shortened.
Author Response
Review of the manuscript ijerph-562898
Reviewer: 1
First, we would like to thank the reviewer for the comments and remarks which have allowed us to improve the paper.
Below we provide a point-to-point reply to the reviewer’s comments, which have been considered in the revised version of the paper.
Reviewer:
As usual with articles whose main novelty lies in exploring a particular dataset, the aspect of discussing results gains major relevance, since otherwise the article’s contribution to the field is only a minor one. Comparison with other environmental indicators in the field is thus required. The discussion need not be lengthy, but some relevant comments are definitely required, so that the plus-values of the GP methodology can be highlighted. A suggestion is to compare, in some meaningful manner, the country rankings of the present article (fig.3) to e.g. those obtained by applying table 1 weights to the same dataset (also, are there any more articles that use table 1 indicators and data? Maybe recheck literature?). A critical analysis of the pros and cons of the proposed GP indicator would also be appreciated. Some comparison GP/EPI is done in L.259-272, but this is insufficient as it only deals with weights, not rankings. Note: this issue is the reason why the review form “Are the conclusions supported by the results?” was answered with “must be improved”.
Authors:
We have compared the country ranking of the GP methodology with the one obtained by applying Table 1 weights from the Environmental Performance Index. This comparison is drawn in Figure 3, which has been updated to cover the ranking of the EPI index. This ranking is represented by points. An explanatory paragraph has been included to perform this comparison:
“Figure 3 also includes the country ranking obtained by applying the weights from Table 1 [9], where scores have been normalized in the range [0,1]. Each country is represented by a single point, in contrast to the GP ranking where lambda parameter enables to obtain a non-crisp value of the environmental performance. The best and worst performer countries in the GP methodology are also consistently ranked in terms of the EPI report. This translates in European countries being in the top of the EPI ranking. However, we can observe a higher variability in those countries located in the middle of the ranking. Finland or Australia were middle performers under the GP methodology, but both of them are highly ranked in the EPI report.
An important conclusion is that most points are on the right side of the GP scores (i.e., on the right side of the boxplot). This means that most countries obtain a high score in the EPI report (58 out of 91 countries are above 50), and it is difficult to draw significant differences between some countries regarding its environmental performance. We have reported that 50% of countries get an EPI score between 39.7 and 71.8, and this makes difficult to highlight significant differences in middle performer countries. In contrast, the ranking obtained through the GP methodology is more balanced, with approximately the same number of countries above and below the level of 50.”
Therefore, the new draft includes the comparison between the rankings obtained through Goal Programming and the weights of Table 1, not only the comparison between the weights.
Other major comments
Reviewer:
In models (4-6) all variables and parameters should be explicitly indicated and their type mentioned (integer, binary, real, range, etc.). This is partly is done after the models are presented, but it would make the formulation more readable if preliminary information concerning the variables and parameters is discriminated beforehand. A suggestion is to follow a structure such as e.g. parameters/variables/objective/constraints, which is common when laying out linear programming models.
Authors:
We have updated models (4-6) in accordance with the referee’ suggestion. We follow the structure proposed by the reviewer, which includes the definition of the parameters and variables at the beginning of the model, and then the objective function and the constraints.
Reviewer:
The role of alpha_j and beta_j in (4-6) should be better explained. “Unwanted” is not enough to understand why they feature in the model. Also, were any alpha/beta = 0 used in the calculations? If so, which one and why?
Authors:
We have included a new paragraph which explain the use of alpha_j and beta_j:
“There are situations where we want to penalize the overachievement of the goals, but not its underachievement; and vice versa. For example, if we are dealing with a cost related variable, we may be interested in controlling the expenses by penalizing the overachievement of the cost goal. However, we may allow for the underachievement of its value.An example can be found in [22].”
Reviewer:
Again in models (4-6), some expressions are redundant. Indeed, the last three lines of these models are just bookkeeping constraints and do not participate in the calculations. They should be set aside from the model definitions or shown in a way similar to what was done in reference [20].
Authors:
We have followed the suggestion made by the reviewer. The models have been shortened by using the structure of the models proposed in reference [20]
Reviewer:
L.172 Values of the expression on line 6 of (4) does NOT coincide with the value of the objective function unless alpha_j and beta_j are 1 for all j. This should be clarified.
Authors:
Following the reviewer’s suggestion, we have introduced a new paragraph to clarify this point:
“We must point out that the sum of D_j may be different from the value obtained in the objective function unless we assume alpha_j = beta_j = 1. Different combinations of values for alpha_j and beta_j serve to perform a trade-off between negative and positive deviations. This would translate into changes regarding the computed weights w.”
Reviewer:
234 The GP sets some weights to (exactly) zero. I have generated and ran some random GP models and verified that zero weights is actually a relatively common occurrence. While this is perfectly fine mathematically, doing away with criteria is not common in multicriteria decision making. Some comment on this particular point is needed. Perhaps correlations play a role here?
Authors:
A new paragraph has been added by following the suggestion made by the reviewer:
“Despite zero weights are usual in GP models [20-22], doing away with criteria is not common in multicriteria decision making. The reason why several criteria are excluded by the GP model is the correlation structure between some criteria. For example, the variable SHI is excluded in all models regardless of the lambda value. A stepwise regression explaining the SHI variable gets an adjusted R-Square of 31.5%, which corresponds to a multiple correlation coefficient of 0.613. Hence, other variables in the sample provide most of the information included in the SHI variable and the GP model discards this variable once those criteria are considered.”
Minor comments
Reviewer:
L.40 and below. Sentences should not start with “[15]” or with a number. Instead of “[15] argues that...” The following form is suggested “In [15] it is argued that...” The same applies to other iterations of the problem during the introduction.
Authors:
According to the reviewer’s suggestion, we have modified all sentences with that structure:
Line 40: “In [15] it is argued that (…)”
Lines 43-44: “All possible preferences among the indicators are considered in [16].”
Lines 52-54: “An analysis of DEA with different combinations of normalization, weighting, and aggregation methods is proposed by [19] for the assessment of an industrial case study on sustainability performance evaluation (…).”
Line 55: “a goal programming model is introduced by [20].”
Reviewer:
L.54 “introduce” -> “introduces”
Authors:
We have changed the paragraph by considering the previous suggestion:
Line 55: “a goal programming model is introduced by [20].”
Reviewer:
L.86 A download link to the dataset would be nice.
Authors:
We have followed the reviewer’s suggestion.
Line 86: “Dataset is available at https://zenodo.org/record/3359779.”
Reviewer:
L.225 Figure 1. Some of the numbers are unperceptible due to light color. Please improve the figure for readability.
Authors:
We have changed Figure 1. The new version draws all numbers in black.
Reviewer:
L.274-277 The paragraph is too long and repetitive. It should be shortened.
Authors:
The paragraph has been shortened:
“The model generated 101 different MEP values per country for $\lambda$ values ranging from 0 to 1 increased by 0.01.”
Reviewer 2 Report
General Comments:
The opening paragraph is overly "climate-change-centric", in my opinion. While climate change is certainly an important environmental issue to consider, it is nowhere near the only driver of environmental performance. And for many of the countries on the list of those evaluated here, I suspect it is nowhere near the most important environmental indicator for them to consider (that is, it is well below local water and air quality factors, which is why they will expend very little resources to address it, perhaps rightly so).
Having said that, I did read this paper with much interest. It is a valuable exercise and does contribute to our understanding of how to properly and meaningfully evaluate environmental performance (if we are trying to aggregate many different factors into one composite "performance indicator"). I do agree with the author that equally weighting all possible factors does not really generate a meaningful aggregated indicator. From a policy perspective, individual countries will certainly have their own weighting mechanisms to determine policy choices.
However, there seems to be a universal desire to aggregate in order to "rank" countries by their environmental performance. As a practical matter, I believe this has fairly little value other than to make the citizens of wealthy, developed countries feel good. In reality, those citizens are largely responsible for the "poor" environmental performance of many developing countries through their consumption of imports produced in those countries, but that is a separate philosophical issue and a separate research question. (For example, if we included "consumption of polluting products" as a national environmental performance indicator, the composite rankings would quite certainly reverse).
Given these individual categories, if we are looking for a useful composite index, this is a step in the right direction. Furthermore, it is an interesting application of the goal programming method. Note that I am an economist and not a mathematician, so these techniques are, in general, new to me and outside my expertise.
I found the paper to be interesting and well written.
Specific comments:
P. 7, lines 210-211 seem to be redundant sentences. Figure 1 is difficult to read. Can this figure (and the accompanying explanation) be made clearer? I understand that these are partial correlation coefficients - can they be made more readable and can statistical significance be added? Perhaps this is just a problem with my printed copy, which was black and white. But is difficult to read in the color version as well.Author Response
Review of the manuscript ijerph-562898
Reviewer: 2
First, we would like to thank the reviewer for the comments and remarks which have allowed us to improve the paper.
Below we provide a point-to-point reply to the reviewer’s comments, which have been considered in the revised version of the paper.
Specific comments:
Reviewer:
7, lines 210-211 seem to be redundant sentences. Figure 1 is difficult to read. Can this figure (and the accompanying explanation) be made clearer? I understand that these are partial correlation coefficients - can they be made more readable and can statistical significance be added? Perhaps this is just a problem with my printed copy, which was black and white. But is difficult to read in the color version as well.
Authors:
We have removed the redundant sentence in lines 201-211. We have also changed Figure 1, and now the correlation coefficients are clearer than in the first manuscript. We have not included the statistical significance because of the size limitations of the figure. Otherwise, the figure would become more difficult to read, and actually the correlation coefficients are not used in the GP methodology (they are solely introduced to give insights regarding how variables relate in the dataset).
I really appreciate one of the points suggested by the reviewer, and I took the liberty of including this consideration in the paper:
“We can conclude that it seems to be a universal desire to aggregate in order to rank countries by their environmental performance. As a practical matter, doing this has fairly little value other than to make the citizens of wealthy, developed countries feel good. In reality, those citizens are largely responsible for the poor environmental performance of many developing countries through their consumption of imports produced in those countries.”